# DNN Feature Map Compression using Learned Representation over GF(2)

## Abstract

In this paper, we introduce a method to compress intermediate feature maps of deep neural networks (DNNs) to decrease memory storage and bandwidth requirements during inference. Unlike previous works, the proposed method is based on converting fixed-point activations into vectors over the smallest GF(2) finite field followed by nonlinear dimensionality reduction (NDR) layers embedded into a DNN. Such an end-to-end learned representation finds more compact feature maps by exploiting quantization redundancies within the fixed-point activations along the channel or spatial dimensions. We apply the proposed network architecture to the tasks of ImageNet classification and PASCAL VOC object detection. Compared to prior approaches, the conducted experiments show a factor of 2 decrease in memory requirements with minor degradation in accuracy while adding only bitwise computations.

## 1 Introduction

Recent achievements of deep neural networks (DNNs) make them an attractive choice in many computer vision applications including image classification (He et al., 2015) and object detection (Huang et al., 2017). The memory and computations required for DNNs can be excessive for low-power deployments. In this paper, we explore the task of minimizing the memory footprint of DNN feature maps during inference and, more specifically, finding a network architecture that uses minimal storage without introducing a considerable amount of additional computations or on-the-fly heuristic encoding-decoding schemes. In general, the task of feature map compression is tightly connected to an input sparsity. The input sparsity can determine several different usage scenarios. This may lead to substantial decrease in memory requirements and overall inference complexity. First, a pen sketches are spatially sparse and can be processed efficiently by recently introduced submanifold sparse CNNs (Graham & van der Maaten, 2017). Second, surveillance cameras with mostly static input contain temporal sparsity that can be addressed by Sigma-Delta networks (O'Connor & Welling, 2017). A more general scenario presumes a dense input e.g. video frames from a high-resolution camera mounted on a moving autonomous car. In this work, we address the latter scenario and concentrate on feature map compression in order to minimize memory footprint and bandwidth during DNN inference which might be prohibitive for high-resolution cameras.

We propose a method to convert intermediate fixed-point feature map activations into vectors over the smallest finite field called the Galois field of two elements (GF(2)) or, simply, binary vectors followed by compression convolutional layers using a nonlinear dimensionality reduction (NDR) technique embedded into DNN architecture. The compressed feature maps can then be projected back to a higher cardinality representation over a fixed-point (integer) field using decompression convolutional layers. Using a layer fusion method, only the compressed feature maps need to be kept for inference while adding only computationally inexpensive bitwise operations. Compression and decompression layers over GF(2) can be repeated within the proposed network architecture and trained in an end-to-end fashion. In brief, the proposed method resembles autoencoder-type (Hinton & Salakhutdinov, 2006) structures embedded into a base network that work over GF(2). Binary conversion and compression-decompression layers are implemented in the Caffe (Jia et al., 2014) framework and publicly available[1].

---

[1]GitHub link will be provided

The rest of the paper is organized as follows. Section 2 reviews related work. Section 3 gives notation for convolutional layers, describes conventional fusion and NDR methods, and explains the proposed method including details about network training and the derived architecture. Section 4 presents experimental results on ImageNet classification and PASCAL VOC object detection using SSD (Liu et al., 2016), memory requirements, and obtained compression rates.

## 2 RELATED WORK

**Feature Map Compression using Quantization.** Unlike a weight compression, surprisingly few papers consider feature map compression. This can most likely be explained by the fact that feature maps have to be compressed for every network input as opposed to offline weight compression. Previous feature map compression methods are primarily developed around the idea of representation *approximation* using a certain quantization scheme: fixed-point quantization (Courbariaux et al., 2015; Gysel et al., 2016), binary quantization (Hubara et al., 2016; Rastegari et al., 2016; Zhou et al., 2016; Tang et al., 2017), and power-of-two quantization (Miyashita et al., 2016). The base floating-point network is converted to the approximate quantized representation and, then, the quantized network is retrained to restore accuracy. Such methods are inherently limited in finding more compact representations since the base architecture remains unchanged. For example, the dynamic fixed-point scheme typically requires around 8-bits of resolution to achieve baseline accuracy for state-of-the-art network architectures. At the same time, binary networks experience significant accuracy drops for large-scale datasets or compact (not over-parametrized) network architectures. Instead, our method can be considered in a narrow sense as a learned quantization using binary representation.

**Embedded NDR Layers**. Another interesting approach is implicitly proposed by Iandola et al. (2016). Although the authors emphasized weight compression rather than feature map compression, they introduced NDR-type layers into network architecture that allowed to decrease not only the number of weights but also feature map sizes by a factor of 8, if one keeps only the outputs of so-called "squeeze" layers. The latter is possible because such network architecture does not introduce any additional convolution *recomputations* since "squeeze" layer computations with a $1 \times 1$ kernel can be fused with the preceding "expand" layers. Our work goes beyond this approach by extending NDR-type layers to work over GF(2) to find a more compact feature map representation.

**Hardware Accelerator Architectures**. Horowitz (2014) estimated that off-chip DRAM access requires approximately $100 \times$ more power than local on-chip cache access. Therefore, currently proposed DNN accelerator architectures propose various schemes to decrease memory footprint and bandwidth. One obvious solution is to keep only a subset of intermediate feature maps at the expense of recomputing convolutions (Alwani et al., 2016). The presented fusion approach seems to be oversimplified but effective due to high memory access cost. Our approach is complementary to this work but proposes to keep only compressed feature maps with minimum additional computations.

Another recent work (Parashar et al., 2017) exploits weight and feature map sparsity using a more efficient encoding for zeros. While this approach targets similar goals, it requires having high sparsity, which is often unavailable in the first and the largest feature maps. In addition, a special control and encoding-decoding logic decrease the benefits of this approach. In our work, compressed feature maps are stored in a dense form without the need of special control and enconding-decoding logic.

## 3 FEATURE MAP COMPRESSION METHODS

### 3.1 MODEL AND NOTATION

The input feature map of $l$th convolutional layer in commonly used DNNs can be represented by a tensor $\mathbf{X}^{l-1} \in \mathbb{R}^{\acute{C} \times H \times W}$, where $\acute{C}$, $H$ and $W$ are the number of input channels, the height and the width, respectively. The input $\mathbf{X}^{l-1}$ is convolved with a weight tensor $\mathbf{W}^l \in \mathbb{R}^{C \times \acute{C} \times H_f \times W_f}$, where $C$ is the number of output channels, $H_f$ and $W_f$ are the height and the width of filter kernel, respectively. A bias vector $\boldsymbol{b} \in \mathbb{R}^C$ is added to the result of convolution operation. Once all $C$ channels are computed, an element-wise nonlinear function is applied to the result of the convolution operations. Then, the $c$th channel of the output tensor $\mathbf{X}^l \in \mathbb{R}^{C \times H \times W}$ can be computed as

$$\mathbf{X}_c^l = g\left(\mathbf{W}_c^l * \mathbf{X}^{l-1} + \boldsymbol{b}_c\right), \tag{1}$$

where $*$ denotes convolution operation and $g()$ is some nonlinear function. In this paper, we assume $g()$ is the most commonly used rectified linear unit (ReLU) defined as $g(x) = \max(0, x)$ such that all activations are non-negative.

## 3.2 CONVENTIONAL METHODS

We formally describe previously proposed methods briefly reviewed in Section 2 using the unified model illustrated in Figure 1. To simplify notation, biases are not shown. Consider a network built using multiple convolutional layers and processed according to (1). Similar to Alwani et al. (2016), calculation of $N$ sequential layers can be fused together without storing intermediate feature maps $\mathbf{X}^{l-N+1}, \ldots, \mathbf{X}^{l-1}$. For example, fusion can be done in a channel-wise fashion using memory buffers which are much smaller than the whole feature map. Then, feature map $\mathbf{X}^l \in \mathbb{R}$ can be quantized into $\hat{\mathbf{X}}^l \in \mathcal{Q}$ using a nonlinear quantization function $q()$ where $\mathcal{Q}$ is a finite field over integers. The quantization step may introduce a drop in accuracy due to imperfect approximation. The network can be further finetuned to restore some of the original accuracy (Courbariaux et al., 2015; Gysel et al., 2016). The network architecture is not changed after quantization and feature maps can be compressed only up to a certain suboptimal bitwidth resolution.

The next step implicitly introduced by Iandola et al. (2016) is to perform NDR using an additional convolutional layer. A mapping $\hat{\mathbf{X}}^l \in \mathcal{Q}^{C \times H \times W} \rightarrow \hat{\mathbf{Y}}^l \in \mathcal{Q}^{\tilde{C} \times H \times W}$ can be performed using projection weights $\mathbf{P}^l \in \mathbb{R}^{\tilde{C} \times C \times H_f \times W_f}$, where the output channel dimension $\tilde{C} < C$. Then, only compressed feature map $\hat{\mathbf{Y}}^l$ needs to be stored in the memory buffer. During the inverse steps, the compressed feature map can be projected back onto the higher-dimensional tensor $\hat{\mathbf{X}}^{l+1} \in \mathcal{Q}$ using weights $\mathbf{R}^l \in \mathbb{R}^{C \times \tilde{C} \times H_f \times W_f}$ and, lastly, converted back to $\mathbf{X}^{l+1} \in \mathbb{R}$ using an inverse quantization function $q^{-1}()$. In the case of a fully quantized network, the inverse quantization can be omitted.

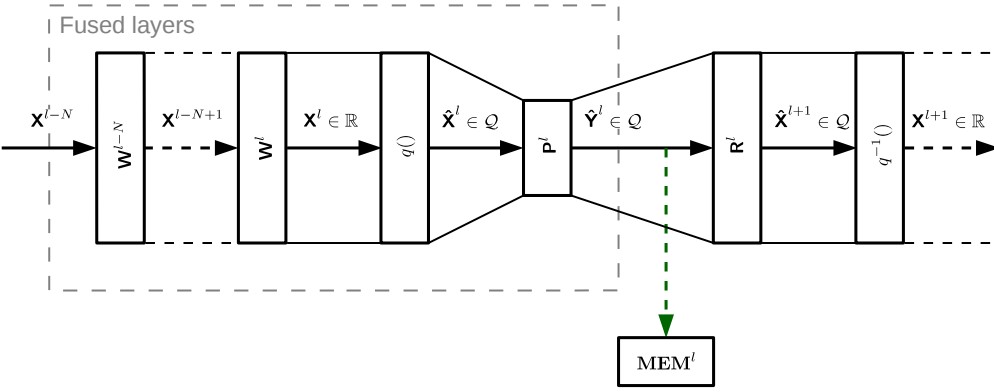

Figure 1: The unified model of conventional methods.

In practice, the number of bits for the feature map quantization step depends on the dataset, network architecture and desired accuracy. For example, over-parameterized architecture like AlexNet may require only 1 or 2 bits for small-scale datasets (CIFAR-10, MNIST, SVHN), but experience significant accuracy drops for large-scale datasets like ImageNet. In particular, the modified AlexNet (with the first and last layers kept in full-precision) top-1 accuracy is degraded by 12.4% and 6.8% for 1-bit XNOR-Net (Rastegari et al., 2016) and 2-bit DoReFa-Net (Zhou et al., 2016), respectively. At the same time, efficient network architectures e.g. Iandola et al. (2016) using NDR layers require 6-8 bits for the fixed-point quantization scheme on ImageNet and fail to work with lower precision activations. In this paper, we follow the path to select an efficient base network architecture and then introduce additional compression layers to obtain smaller feature maps as opposed to initially selecting an over-parametrized network architecture for quantization.

### 3.3 PROPOSED METHOD

#### 3.3.1 REPRESENTATION OVER GF(2)

Consider a scalar $x$ from $\mathbf{X}^l \in \mathbb{R}$. A conventional feature map quantization step can be represented as a scalar-to-scalar mapping or a nonlinear function $\hat{x} = q(x)$ such that

$$x \in \mathbb{R}^{1\times1} \xrightarrow{q()} \hat{x} \in \mathcal{Q}^{1\times1} : \min\|x - \hat{x}\|_2, \tag{2}$$

where $\hat{x}$ is the quantized scalar, $\mathcal{Q}$ is the GF($2^B$) finite field for fixed-point representation and $B$ is the number of bits.

We can introduce a new $\hat{x}$ representation by a linear binarization function $b()$ defined by

$$\hat{x} \in \mathcal{Q}^{1\times1} \xrightarrow{b()} \tilde{\boldsymbol{x}} \in \mathcal{B}^{B\times1} : \tilde{\boldsymbol{x}} = \boldsymbol{b} \otimes \hat{x}, \tag{3}$$

where $\otimes$ is a bitwise AND operation, vector $\boldsymbol{b} = [2^0, 2^1, \ldots, 2^{B-1}]^T$ and $\mathcal{B}$ is GF(2) finite field.

An inverse linear function $b^{-1}()$ can be written as

$$\tilde{\boldsymbol{x}} \in \mathcal{B}^{B\times1} \xrightarrow{b^{-1}()} \hat{x} \in \mathcal{Q}^{1\times1} : \hat{x} = \boldsymbol{b}^T\tilde{\boldsymbol{x}} = \boldsymbol{b}^T\boldsymbol{b} \otimes \hat{x} = (2^B - 1) \otimes \hat{x}. \tag{4}$$

Equations (3)-(4) show that a scalar over a higher cardinality finite field can be linearly converted to and from a vector over a finite field with two elements. Based on these derivations, we propose a feature map compression method shown in Figure 2. Similar to (Courbariaux et al., 2015), we quantize activations to obtain $\hat{\mathbf{X}}^l$ and, then, apply transformation (3). The resulting feature map can be represented as $\tilde{\mathbf{X}}^l \in \mathcal{B}^{B\times C\times H\times W}$. For implementation convenience, a new bit dimension can be concatenated along channel dimension resulting in the feature map $\tilde{\mathbf{X}}^l \in \mathcal{B}^{BC\times H\times W}$. Next, a single convolutional layer using weights $\mathbf{P}^l$ or a sequence of layers with $\mathbf{P}^l_i$ weights can be applied to obtain a compressed representation over GF(2). Using the fusion technique, only the compressed feature maps $\tilde{\mathbf{Y}}^l \in \mathcal{B}$ need to be stored in memory during inference. Non-compressed feature maps can be processed using small buffers e.g. in a sequential channel-wise fashion. Lastly, the inverse function $b^{-1}()$ from (4) using convolutional layers $\mathbf{R}^l_i$ and inverse of quantization $q^{-1}()$ undo the compression and quantization steps.

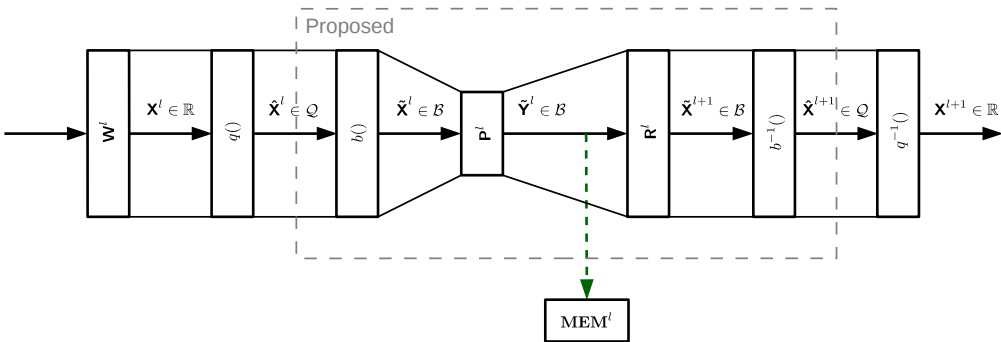

Figure 2: Scheme of the proposed method.

#### 3.3.2 LEARNING OVER GF(2)

The graph model shown in Figure 3 explains details about the inference (forward pass) and backpropagation (backward pass) phases of the newly introduced functions. The inference phase represents (3)-(4) as explained above.

Clearly, the backpropagation phase may seem not obvious at a first glance. One difficulty related to the quantized network is that quantization function itself is not differentiable. But many studies e.g. Courbariaux et al. (2015) show that a mini-batch-averaged floating-point gradient practically

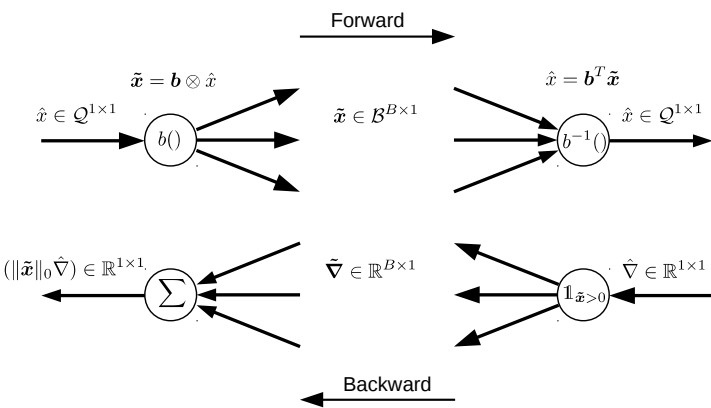

Figure 3: Forward and Backward passes during inference and backpropagation.

works well assuming quantized forward pass. The new functions $b()$ and $b^{-1}()$ can be represented as gates that make hard decisions similar to ReLU. The gradient of $b^{-1}()$ can then be calculated using results of Bengio et al. (2013) as

$$\hat{\nabla} \in \mathbb{R}^{1 \times 1} \xrightarrow{b^{-1}()} \tilde{\nabla} \in \mathbb{R}^{B \times 1} : \tilde{\nabla} = \mathbb{1}_{\tilde{x}>0} \nabla. \tag{5}$$

Lastly, the gradient of $b()$ is just a scaled sum of the gradient vector calculated by

$$\tilde{\nabla} \in \mathbb{R}^{B \times 1} \xrightarrow{b()} \hat{\nabla} \in \mathbb{R}^{1 \times 1} : \hat{\nabla} = \mathbb{1}^T \tilde{\nabla} = \mathbb{1}^T \mathbb{1}_{\tilde{x}>0} \nabla = \|\tilde{x}\|_0 \nabla, \tag{6}$$

where $\|\tilde{x}\|_0$ is a gradient scaling factor that represents the number of nonzero elements in $\tilde{x}$. Practically, the scaling factor can be calculated based on statistical information only once and used as a static hyperparameter for gradient normalization.

Since the purpose of the network is to learn and keep only the smallest $\tilde{Y}^l$, the choice of $\mathbf{P}^l$ and $\mathbf{R}^l$ initialization is important. Therefore, we can initialize these weight tensors by an identity function that maps the non-compressed feature map to a truncated compressed feature map and vice versa. That provides a good starting point for training. At the same time, other initializations are possible e.g. noise sampled from some distribution studied by Bengio et al. (2013) can be added as well.

### 3.3.3 NETWORK ARCHITECTURE

To highlight benefits of the proposed approach, we select a base network architecture with the compressed feature maps according to Section 3.2. The selected architecture is based on a quantized SqueezeNet network. It consists of a sequence of "fire" modules where each module contains two concatenated "expand" layers and a "squeeze" layer illustrated in Figure 4. The "squeeze" layers perform NDR over the field of real or, in case of the quantized model, integer numbers. Specifically, the size of concatenated "expand 1×1" and "expand 3×3" layers is compressed by a factor of 8 along channel dimension by "squeeze 1×1" layer. Activations of only the former one can be stored during inference according to the fusion method. According to the analysis presented in Gysel et al. (2016), activations quantized to 8-bit integers do not experience significant accuracy drop.

The given base architecture is extended by the proposed method. The quantized "squeeze" layer feature map is converted to its binary representation following Figure 2. Then, the additional compression rate is defined by selecting parameters of $\mathbf{P}^l_i$. In the simplest case, only a single NDR layer can be introduced with the weights $\mathbf{P}^l$. In general, a number of NDR layers can be added with 1×1, 3×3 and other kernels with or without pooling at the expense of increased computational cost. For example, 1×1 kernels allow to learn optimal quantization and to compensate redundancies along channel dimension only. But 3×3 kernels can address spatial redundancies and, while being implemented with stride 2 using convolutional-deconvolutional layers, decrease feature map size along spatial dimensions.

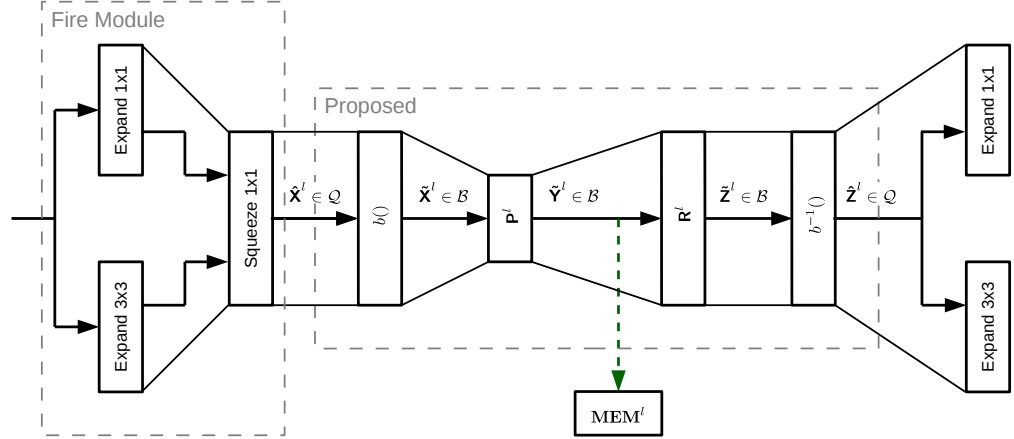

Figure 4: Extended "fire" module of SqueezeNet by the proposed method.

# 4 EXPERIMENTS

## 4.1 IMAGENET CLASSIFICATION

We implemented the new binarization layers from Section 3 as well as quantization layers using modified (Gysel et al., 2016) code in the Caffe (Jia et al., 2014) framework. The latter code is modified to accurately support binary quantization during inference and training. SqueezeNet v1.1 is selected as a base floating-point network architecture, and its pretrained weights were downloaded from the publicly available source[2]. In the experiments, we compress the "fire2/squeeze" and "fire3/squeeze" layers which are the largest of the "squeeze" layers due to high spatial dimensions. The input to the network has a resolution of 227×227, and the weights are all floating-point.

Table 1: ImageNet accuracy: A - "fire2,3/squeeze" feature maps and W - weights.

| Model | W size, MB | A size, KB | Speed, FPS | Top-1 Accuracy, % | Top-5 Accuracy, % |
|---|---|---|---|---|---|
| fp32 | 4.7 | 392.0 | 298 | 58.4 | 81.0 |
| Quantized | | | | | |
| uint8 | 4.7 | 98.0 | 298 | 58.6(58.3) | 81.1(81.0) |
| uint6 | 4.7 | 73.5 | 298 | 57.8(55.5) | 80.7(78.7) |
| uint4 | 4.7 | 49.0 | 298 | 54.9(18.0) | 78.3(34.2) |
| Proposed: $b()$ →1×1→1×1→ $b^{-1}()$ | | | | | |
| uint6 | 5.0 | 73.5 | 244 | 58.8 | 81.3 |
| uint4 | 4.9 | 49.0 | 244 | 57.3 | 80.0 |
| Proposed: $b()$ →3×3/2→3×3*2→ $b^{-1}()$ | | | | | |
| uint8 | 7.6 | 24.5 | 223 | 54.1 | 77.4 |
| uint6 | 6.9 | 18.4 | 223 | 53.8 | 77.2 |

The quantized and compressed models are retrained for 100,000 iterations with a mini-batch size of 1024 on the ImageNet (Russakovsky et al., 2015) (ILSVRC2012) training dataset, and SGD solver with a step-policy learning rate starting from 1e-3 divided by 10 every 20,000 iterations. Although this large mini-batch size was used by the original model, it helps the quantized and compressed models to estimate gradients as well. The compressed models were derived and retrained iteratively from the 8-bit quantized model. Table 1 reports top-1 and top-5 inference accuracies of 50,000 images from ImageNet validation dataset. The fourth column in Table 1 represents speed in terms

---

[2]https://github.com/DeepScale/SqueezeNet

of frames per second (FPS) of network forward pass with mini-batch size of 1 using NVIDIA Tesla P100 GPU. All quantized computations are emulated on GPU using fp32. Hence, it shows a measure of emulation speed for additional layers rather than speed of the optimized layers.

According to Table 1, the quantized models after retraining experience -0.2%, 0.6% and 3.5% top-1 accuracy drops for 8-bit, 6-bit and 4-bit quantization, respectively. For comparison, the quantized models without retraining are shown in parentheses. The proposed compression method using $1 \times 1$ kernels allows us to restore corresponding top-1 accuracy by 1.0% and 2.4% for 6-bit and 4-bit versions at the expense of a small increase in the number of weights and bitwise convolutions. Moreover, we evaluated a model with a convolutional layer followed by a deconvolutional layer both with a $3 \times 3$ stride 2 kernel at the expense of a 47% increase in weight size for 6-bit activations. That allowed us to decrease feature maps in spatial dimensions by exploiting local spatial quantization redundancies. Then, the feature map size is further reduced by a factor of 4, while top-1 accuracy dropped by 4.3% and 4.6% for 8-bit and 6-bit activations, respectively. A comprehensive comparison with the state-of-the-art networks is given in Appendix A.

## 4.2 PASCAL VOC OBJECT DETECTION USING SSD

**Accuracy experiments**. We evaluate object detection using Pascal VOC (Everingham et al., 2010) dataset which is a more realistic application for autonomous cars where the high-resolution cameras emphasize feature map compression benefits. The VOC2007 test dataset contains 4,952 images and a training dataset of 16,551 images is a union of VOC2007 and VOC2012. We adopted SSD512 model (Liu et al., 2016) for the proposed architecture. SqueezeNet pretrained on ImageNet is used as a feature extractor instead of the original VGG-16 network. This reduces number of parameters and overall inference time by a factor of 4 and 3, respectively. The original VOC images are rescaled to 512×512 resolution. As with ImageNet experiments, we generated several models for comparisons: a base floating-point model, quantized models, and compressed models. We apply quantization and compression to the "fire2/squeeze" and "fire3/squeeze" layers which represent, if the fusion technique is applied, more than 80% of total feature map memory due to their large spatial dimensions. Typically, spatial dimensions decrease quadratically because of max pooling layers compared to linear growth in the depth dimension. The compressed models are derived from the 8-bit quantized model, and both are retrained for 10,000 mini-batch-256 iterations using SGD solver with a step-policy learning rate starting from 1e-3 divided by 10 every 2,500 iterations.

Table 2: VOC2007 SSD512 accuracy: A - "fire2,3/squeeze" feature maps and W - weights.

| Model | W size, MB | A size, KB | Speed, FPS | mAP, % |
|---|---|---|---|---|
| VGG-16, fp32 | 103.7 | - | 34 | 76.80 |
| SqueezeNet, fp32 | 23.7 | 2048 | 104 | 68.12 |
| Quantized | | | | |
| SqueezeNet, uint8 | 23.7 | 512 | 102 | 68.08(68.04) |
| SqueezeNet, uint6 | 23.7 | 384 | 102 | 67.66(67.14) |
| SqueezeNet, uint4 | 23.7 | 256 | 102 | 65.92(44.13) |
| SqueezeNet, uint2 | 23.7 | 128 | 102 | 55.86(0.0) |
| Proposed $b() \rightarrow 1\times1 \rightarrow 1\times1 \rightarrow b^{-1}()$ | | | | |
| SqueezeNet, uint6 | 23.9 | 384 | 86 | 68.17 |
| SqueezeNet, uint4 | 23.8 | 256 | 86 | 65.42 |
| Proposed: $b() \rightarrow 3\times3/2 \rightarrow 3\times3*2 \rightarrow b^{-1}()$ | | | | |
| SqueezeNet, uint8 | 26.5 | 128 | 78 | 63.53 |
| SqueezeNet, uint6 | 25.9 | 96 | 78 | 62.22 |
| Proposed: $b() \rightarrow 2\times2/2 \rightarrow 2\times2*2 \rightarrow b^{-1}()$ | | | | |
| SqueezeNet, uint8 | 24.9 | 128 | 82 | 64.39 |
| SqueezeNet, uint6 | 24.6 | 96 | 82 | 62.09 |

Table 2 presents mean average precision (mAP) results for the original VGG-16 and SqueezeNet-based models as well as size of the weights, feature maps to compress and the speed metric with

Section 4.1 assumptions. The 8-bit quantized model with retraining drops accuracy by less than 0.04%, while 6-bit, 4-bit and 2-bit models decrease accuracy by 0.5%, 2.2% and 12.3%, respectively. For reference, mAPs for the quantized models without retraining are shown in parentheses. Using the proposed compression-decompression layers with a $1{\times}1$ kernel, mAP for the 6-bit model is increased by 0.5% and mAP for the 4-bit is decreased by 0.5%. We conclude that compression along channel dimension is not beneficial for SSD unlike ImageNet classification either due to low quantization redundancy in that dimension or the choice of hyper-parameters e.g. mini-batch size. Then, we evaluate the models with spatial-dimension compression which is intuitively appealing for high-resolution images. Empirically, we found that a $2{\times}2$ kernel with stride 2 performs better than a corresponding $3{\times}3$ kernel while requiring less parameters and computations. According to Table 2, an 8-bit model with $2{\times}2$ kernel and downsampling-upsampling layers achieves 1% higher mAP than a model with $3{\times}3$ kernel and only 3.7% lower than the base floating-point model.

**Memory Requirements**. Table 3 summarizes memory footprint benefits for the evaluated SSD models. Similar to the previous section, we consider only the largest feature maps that represent more than 80% of total activation memory. Assuming that the input frame is stored separately, the fusion technique allows to compress feature maps by a factor of 19. Note that no additional recomputations are needed. Second, conventional 8-bit and 4-bit fixed-point models decrease the size of feature maps by a factor of 4 and 8, respectively. Third, the proposed model with $2{\times}2$ stride 2 kernel gains another factor of 2 compression compared to 4-bit fixed-point model with only 1.5% degradation in mAP. This result is similar to ImageNet experiments which showed relatively limited compression gain along channel dimension only. At the same time, learned quantization along combined channel and spatial dimensions pushes further compression gain. In total, the memory footprint for this feature extractor is reduced by two orders of magnitude.

Table 3: Memory requirements: F - fusion, Q - quantization, C - compression ($2{\times}2$/2).

| Layer / Size, KB | Activation Dimensions | Base, fp32 | F, fp32 | F+Q, uint8 | F+Q, uint4 | F+Q+C, uint8 |
|---|---|---|---|---|---|---|
| input (int8) | $3{\times}512{\times}512$ | 768 | 768 | 768 | 768 | 768 |
| conv1 | $64{\times}256{\times}256$ | 16384 | 0 | 0 | 0 | 0 |
| maxpool1 | $64{\times}128{\times}128$ | 4096 | 0 | 0 | 0 | 0 |
| fire2,3/squeeze | $2{\times}16{\times}128{\times}128$ | 2048 | 2048 | 512 | 256 | 128 |
| fire2,3/expand | $2{\times}128{\times}128{\times}128$ | 16384 | 0 | 0 | 0 | 0 |
| Total, KB | | 38912 | 2048 | 512 | 256 | 128 |
| mAP, % | | 68.12 | 68.12 | 68.08 | 65.92 | 64.39 |
| Compression | | - | 19$\times$ | 76$\times$ | 152$\times$ | 304$\times$ |

## 5 CONCLUSIONS

We introduced a method to decrease memory storage and bandwidth requirements for DNNs. Complementary to conventional approaches that use fused layer computation and quantization, we presented an end-to-end method for learning feature map representations over GF(2) within DNNs. Such a binary representation allowed us to compress network feature maps in a higher-dimensional space using autoencoder-inspired layers embedded into a DNN along channel and spatial dimensions. These compression-decompression layers can be implemented using conventional convolutional layers with bitwise operations. To be more precise, the proposed representation traded cardinality of the finite field with the dimensionality of the vector space which makes possible to learn features at the binary level. The evaluated compression strategy for inference can be adopted for GPUs, CPUs or custom accelerators. Alternatively, existing binary networks can be extended to achieve higher accuracy for emerging applications such as object detection and others.

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

## A    COMPARISON WITH THE STATE-OF-THE-ART NETWORKS

We compare recently reported ImageNet results for networks that compress feature maps as well as several configurations of the proposed approach. Most of the works use the over-parametrized AlexNet architecture while ours is based on SqueezeNet. Table 4 shows accuracy results for base networks as well as their quantized versions. Binary XNOR-Net (Rastegari et al., 2016) estimates based on AlexNet as well as ResNet-18. DoReFa-Net (Zhou et al., 2016) is more flexible and can adjust the number of bits for weights and activations. Since its accuracy is limited by the number of activation bits, we present three cases with 1-bit, 2-bit, and 4-bit activations. The most recent work (Tang et al., 2017) solves the problem of binarizing the last layer weights, but weights of the first layer are full-precision. Overall, AlexNet-based low-precision networks achieve 43.6%, 49.8%, 53.0% top-1 accuracy for 1-bit, 2-bit and 4-bit activations, respectively. Around 70% of the memory footprint is defined by the first two layers of AlexNet. The fusion technique is difficult in such architectures due to large kernel sizes ($11 \times 11$ and $5 \times 5$ for AlexNet) which can cause extra recomputations. Thus, activations require 95.4KB, 190.0KB and 381.7KB of memory for 1-bit, 2-bit and 4-bit models, respectively. The NIN-based network from (Tang et al., 2017) with 2-bit activations achieves 51.4% top-1 accuracy, but its activation memory is larger than AlexNet due to late pooling layers.

Table 4: ImageNet accuracy: W - weights, A - feature maps.

| Methods | Base Model | W, bits | W size, MB | A, bits | A size, KB | Top-1 Accuracy, % | Top-5 Accuracy, % |
|---|---|---|---|---|---|---|---|
| AlexNet | - | 32 | 232 | 32 | 3053.7 | 56.6 | 79.8 |
| AlexNet | - | 32 | 232 | 6 | 572.6 | 55.8 | 79.2 |
| XNOR-Net | AlexNet | $1^1$ | 22.6 | 1 | $344.4^2$ | 44.2 | 69.2 |
| XNOR-Net | ResNet-18 | $1^1$ | 3.34 | 1 | $1033.0^2$ | 51.2 | 73.2 |
| DoReFa-Net | AlexNet | $1^1$ | 22.6 | 1 | 95.4 | 43.6 | - |
| DoReFa-Net | AlexNet | $1^1$ | 22.6 | 2 | 190.9 | 49.8 | - |
| DoReFa-Net | AlexNet | $1^1$ | 22.6 | 4 | 381.7 | 53.0 | - |
| Tang'17 | AlexNet | $1^3$ | 7.43 | 2 | 190.9 | 46.6 | 71.1 |
| Tang'17 | NIN-Net | $1^3$ | 1.23 | 2 | 498.6 | 51.4 | 75.6 |
| The proposed models | | | | | | | |
| SqueezeNet | - | 32 | 4.7 | 32 | 12165.4 | 58.4 | 81.0 |
| F+Q | SqueezeNet | 8 | 1.2 | 8 | 189.9 | 58.3 | 80.8 |
| F+Q+C($1 \times 1$) | SqueezeNet | 8 | 1.2 | $6(8)^4$ | 165.4 | $58.3(58.8)^5$ | $81.0(81.3)^5$ |
| F+Q+C($1 \times 1$) | SqueezeNet | 8 | 1.2 | $4(8)^4$ | 140.9 | $56.6(57.3)^5$ | $79.7(80.0)^5$ |
| F+Q+C($3 \times 3s2$) | SqueezeNet | 8 | 1.9 | $8(8)^4$ | 116.4 | $53.5(54.1)^5$ | $76.7(77.4)^5$ |
| F+Q+C($3 \times 3s2$) | SqueezeNet | 8 | 1.7 | $6(8)^4$ | 110.3 | $53.0(53.8)^5$ | $76.8(77.2)^5$ |

[1] Weights are not binarized for the first and the last layer.
[2] Activations estimates are based on 8-bit assumption since it is not clear from Rastegari et al. (2016) whether the activations were binarized or not for the first and the last layer.
[3] Weights are not binarized for the first layer.
[4] Number of bits for the compressed "fire2,3/squeeze" layers and, in parentheses, for quantized only layers.
[5] For comparison, accuracy in parentheses represents result for the corresponding model in Table 1.

The SqueezeNet-based models in Table 4 are finetuned from the corresponding models in Table 1 for 40,000 iterations with a mini-batch size of 1024, and SGD solver with a step-policy learning rate starting from 1e-4 divided by 10 every 10,000 iterations. The model with fusion and 8-bit quantized weights and activations, while having an accuracy similar to floating-point model, outperforms the state-of-the-art networks in terms of weight and activation memory. The proposed four models from Table 1 further decrease activation memory by adding compression-decompression layers to "fire2,3/squeeze" modules. This step allowed to shrink memory from 189.9KB to 165.4KB, 140.9KB, 116.4KB and 110.3KB depending on the compression configuration. More compression is possible, if apply the proposed approach to other "squeeze" layers.

