# OpenReview forum: "DNN Feature Map Compression using Learned Representation over GF(2)"
_ICLR.cc/2018/Conference — Reject_

### Official Review · AnonReviewer2 · 2017-11-24
**initial review**

**Rating:** 7
**Confidence:** 3

**Review:**

The method of this paper minimizes the memory usage of the activation maps of a CNN. It starts from a representation where activations are compressed with a uniform scalar quantizer and fused to reduce intermediate memory usage. This looses some accuracy, so the contribution of the paper is to add a pair of convolution layers in the binary domain (GF(2)) that are trained to restore the lost precision.

Overall, this paper seems to be a nice addition to the body of works on network compression.

+ : interesting approach and effective results.

+ : well related to the state of the art and good comparison with other works.

- : somewhat incremental. Most of the claimed 100x compression is due to previous work.

- : impact on runtime is not reported. Since there is a caffe implementation it would be interesting to have an additional column with the comparative execution speeds, even if only on CPU. I would expect the FP32 timings to be hard to beat, despite the claims that it uses only binary operations.

- : the paper is sometimes difficult to understand (see below)

detailed comments:

Equations (3)-(4) are difficult to understand. If I understand correctly, b just decomposes a \hat{x} in {0..2^B-1} into its B bits \tilda{x} \in {0,1}^B, which can be then considered as an additional dimension in the activation map where \hat{x} comes from.

It is not stated clearly whether P^l and R^l have binary weights. My understanding is that P^l has but R^l not.

4.1 --> a discussion of the large mini-batch size (1024) could be useful. My understanding is that large mini-batches are required to use averaged gradients and get smooth updates.

end of 4.1 --> unclear what "equivalent bits" means

---

> ### Author Response · Authors · 2017-12-16
> **to AnonReviewer2**
>
> > Overall, this paper seems to be a nice addition to the body of works on network compression.
>
> Thank you.
>
> - : somewhat incremental. Most of the claimed 100x compression is due to previous work.
>
> We agree that compressing along channel dimension improved compression by relatively small amount (~1/3) with comparable accuracy. At the same time:
> a) We selected state-of-the-art architecture which is hard to compress unlike general unoptimized networks.
> b) Practically, even ~1/3 improvement may lead to significant improvements if off-chip bandwidth can be completely avoided.
> c) According to Table 1, the scheme with convolutional-deconvolutional layers compressed feature maps by another factor of 4 which is significant improvement.
> d) Due to lack of time/space we didn't experiment with multiple compression layers which is another option.
>
> Unfortunately, c) didn't work well for Faster R-CNN. We believe, it is because training of the Faster R-CNN is limited to batch size = 1. Currently, we try to get results for SSD detector which doesn't have such limitation.
>
> - : impact on runtime is not reported. Since there is a caffe implementation it would be interesting to have an additional column with the comparative execution speeds, even if only on CPU. I would expect the FP32 timings to be hard to beat, despite the claims that it uses only binary operations.
>
> We have both CPU and GPU implementations and added speed numbers to Table 1. However, these numbers represent emulation speed because new quantization and compression layers are emulated on GPU in fp32 rather than efficiently processed. So, these numbers measure relative overhead for emulation of quantization and compression layers.
>
> - : the paper is sometimes difficult to understand (see below)
>
> detailed comments:
>
> - : Equations (3)-(4) are difficult to understand. If I understand correctly, b just decomposes a \hat{x} in {0..2^B-1} into its B bits \tilda{x} \in {0,1}^B, which can be then considered as an additional dimension in the activation map where \hat{x} comes from.
>
> Correct. We preferred a more formal and compact description to save some space.
>
> - : It is not stated clearly whether P^l and R^l have binary weights. My understanding is that P^l has but R^l not.
>
> We do not consider weight quantization in this paper. All weights are floating-point from notation given in 3.2 and as stated in the 1st paragraph of 4.1. The only exception is Appendix A where weights are 8-bit integers to show benefits of optimized architecture compared to binarized networks in terms of weight size. We added some missing notation in the 2nd paragraph of Section 3.2 as well.
>
> - : 4.1 --> a discussion of the large mini-batch size (1024) could be useful. My understanding is that large mini-batches are required to use averaged gradients and get smooth updates.
>
> Thank you, we added this discussion to 4.1 and 4.2. We used the original mini-batch size from SqueezeNet authors for image classification. To be precise, they set global mini-batch size = mini-batch_size * iter_size = 32 * 32 = 1024 in Caffe. So, global mini-batch size of 1024 is achieved by using 32 iterations each of size 32. Hence, large mini-batch is used to train such optimized architecture even in fp32. We agree that the large mini-batch allows to smooth quantization effects as well.
>
> - : end of 4.1 --> unclear what "equivalent bits" means
>
> Thank you. We removed this to not to confuse readers and free some space. The idea was to differentiate between bits for binary vectors and integers.

---

### Official Review · AnonReviewer3 · 2017-11-28
**DNN Feature Map Compression using Learned Representation**

**Rating:** 5
**Confidence:** 4

**Review:**

In order to compress DNN intermediate feature maps the authors covert fixed-point activations into vectors over the smallest finite field, the Galois field of two elements (GF(2)) and use nonlinear dimentionality reduction layers.

The paper reads well and the methods and experiments are generally described in sufficient detail.

My main concern with this paper and approach is the performance achieved. According to Table 1 and Table 2 there is a small accuracy benefit from using the proposed approach over the "quantized" SqueezeNet baseline. If I am weighing in the need to alter the network for the proposed approach in comparison with the "quantized" setting then, from practical point of view, I would prefer the later "quantized" approach.

---

> ### Author Response · Authors · 2017-12-16
> **to AnonReviewer3**
>
> - The paper reads well and the methods and experiments are generally described in sufficient detail.
>
> Thank you.
>
> > My main concern with this paper and approach is the performance achieved. According to Table 1 and Table 2 there is a small accuracy benefit from using the proposed approach over the "quantized" SqueezeNet baseline. If I am weighing in the need to alter the network for the proposed approach in comparison with the "quantized" setting then, from practical point of view, I would prefer the later "quantized" approach.
>
> We agree that compressing along channel dimension improved compression by relatively small amount (~1/3) with comparable accuracy. At the same time:
> a) We selected state-of-the-art architecture which is hard to compress unlike general unoptimized networks.
> b) Practically, even ~1/3 improvement may lead to significant improvements if off-chip bandwidth can be completely avoided.
> c) According to Table 1, the scheme with convolutional-deconvolutional layers compressed feature maps by another factor of 4 which is significant improvement.
> d) Due to lack of time/space we didn't experiment with multiple compression layers which is another option.
>
> Unfortunately, c) didn't work well for Faster R-CNN. We believe, it is because training of the Faster R-CNN is limited to batch size = 1. Currently, we try to get results for SSD which doesn't have such limitation.

---

### Official Review · AnonReviewer1 · 2017-11-28
**high compression rate for marginal accuracy loss, but approach requires specialized architecture**

**Rating:** 4
**Confidence:** 4

**Review:**

Strengths:
- Unlike most previous approaches that suffer from significant accuracy drops for good feature map compression, the proposed method achieves reductions in feature map sizes of 1 order of magnitude at effectively no loss in accuracy.
- Technical approach relates closely to some of the prior approaches (e.g., Iandola et al. 2016) but can be viewed as learning the quantization rather than relying on a predefined one.
- Good results on both large-scale classification and object detection.
- Technical approach is clearly presented.

Weaknesses:
- The primary downside is that the approach requires a specialized architecture to work well (all experiments are done with SqueezeNets). Thus, the approach is less general than prior work, which can be applied to arbitrary architectures.
- From the experiments it is not fully clear what is the performance loss due to having to use the SqueezeNet architecture rather than state-of-the-art models. For example, for the image categorization experiment, the comparative baselines are for AlexNet and NIN, which are outdated and do not represent the state-of-the-art in this field. The object detection experiments are based on a variant of Faster R-CNN where the VGG16 feature extractor is replaced with a SqueezeNet model. However, the drop in accuracy caused by this modification is not discussed in the paper and, in any case, there are now much better models for object detection than Faster R-CNN.
- In my view the strengths of the approach would be more convincingly conveyed visually with a plot reporting accuracy versus memory usage, rather than by the many numerical tables in the paper.

---

> ### Author Response · Authors · 2017-12-16
> **to AnonReviewer1**
>
> - The primary downside is that the approach requires a specialized architecture to work well (all experiments are done with SqueezeNets). Thus, the approach is less general than prior work, which can be applied to arbitrary architectures.
>
> We selected SqueezeNet just because it is state-of-the-art in terms of feature map size (for non- binary/ternary networks). Then, we clarified previously unpublished aspects of this network when combined with the fusion approach, proposed unified model of feature map compression used in SqueezeNet and our method, and showed improvements compared to prior work. The same compression strategy can be applied to any network architecture by introducing compression layers.
>
> - From the experiments it is not fully clear what is the performance loss due to having to use the SqueezeNet architecture rather than state-of-the-art models. For example, for the image categorization experiment, the comparative baselines are for AlexNet and NIN, which are outdated and do not represent the state-of-the-art in this field.
>
> The only competitive prior work in terms of feature map footprint are binary/ternary networks. Unfortunately, binary/ternary networks work well only for over-parametrized networks like AlexNet. We report ResNet-18 and NiN as well. These are the only published results we could find to compare to. We couldn't find reported binary/ternary networks derived from ImageNet state-of-the-art networks for the above reasons.
>
> - The object detection experiments are based on a variant of Faster R-CNN where the VGG16 feature extractor is replaced with a SqueezeNet model. However, the drop in accuracy caused by this modification is not discussed in the paper and, in any case, there are now much better models for object detection than Faster R-CNN.
>
> Thank you, we added numbers for VGG-16 Faster R-CNN. Could you clarify what are much better models for object detection? We believe, Faster R-CNN, R-FCN, SSD and YOLO are the most popular approaches. For example, recent CVPR17 paper(https://arxiv.org/abs/1611.10012) accomplished comprehensive comparisons and showed that Faster R-CNN might be better in terms of speed/accuracy than others.
> Now we try to add results for SSD detector as well. We believe, that SSD would give us a better result than Faster R-CNN because training of the latter is limited to batch size = 1.
>
> - In my view the strengths of the approach would be more convincingly conveyed visually with a plot reporting accuracy versus memory usage, rather than by the many numerical tables in the paper.
>
> We tried. Unfortunately, the linear scale doesn't look good in such plots due to big gaps in memory size numbers. We will try to make Tables more elegant looking.

---

### Author Response · Authors · 2017-12-24
**Major revision**

Dear reviewers, we went through a major revision. Updates:
1. We replaced Faster R-CNN with SSD detector. As we presumed, Faster R-CNN's hard limitation on mini-batch size = 1 was a limiting factor for downsampling-upsampling compression layers. Thanks to R2 who pointed us to this direction.

2. The main concern of R2/R3 was the achieved compression gain. We agreed that the compression along channel dimension only was limited and, hence, concentrated on a more promising approach with downsampling-upsampling layers which allow to learn quantization redundancies along combined channel and local spatial dimensions. In the 1st revision of the paper we could get results with spatial-dimension compression on ImageNet classifier but was not able to do this for object detector due to #1. After switching to SSD and setting a reasonable mini-batch size of 256, we could apply such layers for detector as well. According to Table 1 and 2, this approach provides additional ~2x compression gain compared to prior works with minor accuracy degradation.

3. Section 4.2/4.3 was rewritten to reflect #1 and #2 changes.

4. Empirically, we found that 2x2 kernel with stride 2 works better (higher accuracy and less extra parameters) for SSD.

---

### Decision · Program_Chairs · 2018-01-29
**ICLR 2018 Conference Acceptance Decision**

**Decision:**

Reject

**Comment:**

The paper presents a technique for feature map compression at inference time. As noted by reviewers, the main concern is that the method is applied to one NN architecture (SqueezeNet), which severely limits its impact and applicability to better performing state-of-the-art models.